# Action Schema Networks for Numeric Planning

## Afifa Tariq, Richard Valenzano, Mikhail Soutchanski

Toronto Metropolitan University, Ontario, Canada
afifa.tariq@torontomu.ca, rick.valenzano@torontomu.ca, mes@cs.torontomu.ca

## Abstract

Planning is the fundamental ability of an intelligent agent to reason about what decisions it should make in a given environment to achieve a particular set of goals. Generalized planning is the task of finding a generalized policy that applies to a set of planning instances that share a standard model. Action Schema Networks (ASNets) is an approach to find generalized policies for classical planning problems. In this paper, we extend ASNet to work with numeric planning problems. We use a technique to propositionalize numeric variables, which converts them from infinite ranges to a finite domain, and update the training procedure to use exploration in order to increase the diversity of states encountered. We also use a non-generalized numeric planner, Expressive Numeric Heuristic Search Planner (ENHSP), to teach ASNet to solve numeric planning problems by learning to mimic the actions chosen by ENHSP for problem instances. ASNet finds a generalized policy and weights after training, allowing it to share these to solve unseen problem instances of the same domain. We analyze our approach through an extensive experimental study aimed at evaluating the performance of ASNet on several numeric planning domains. The results show that our numeric ASNet can effectively solve problems in many numeric planning domains.

## Introduction

*Planning* is the process of identifying a sequence of actions in a given environment to achieve a particular set of goals. In *classical planning* problems, the states of the environment are represented using propositions of finite domain variables. However, many complex real-world problems involving resource consumption or other numeric features which cannot adequately be represented using propositional variables. This led to the introduction of *numeric planning*, which does so using numeric variables.

Much of the research on numeric planning has been done by extending the techniques used for classical planning to apply to numeric problems. This is the approach that we have used in this paper, as we extend *Action Schema Networks (ASNet)* (Toyer et al. 2018, 2020) — which is a state-of-the-art approach and the first standard neural network architecture designed for generalized classical planning problems — to work with numeric planning problems.

*Generalized planning* is the task of finding a general policy (*i.e.* a mapping of states to actions) that is applicable to a set of planning instances that share a standard model. With our extension of ASNet, we have built a generalized planner, which uses neural networks to learn a generalized policy for numeric planning from training examples.

An ASNet is a neural network which takes in features corresponding to actions and propositions from a planning problem and outputs values for each applicable action in the current state. The main intuition behind ASNet is for the neural network to learn to mimic the actions chosen by a non-generalized planner on some training problems for the given domain. An appropriately-learnt set of weights can then be used to obtain a generalized policy. In particular, an ASNet is created for each problem instance of a domain by sharing neural network weights to solve other problem instances of the same domain.

In this work, we use a simple approach to convert a numeric planning problem into a classical planning problem. We do so by grounding the numeric propositions of the problem by assigning them values from a fixed range of values. We also enhance the training algorithm to better suit numeric planning problems. We also add an exploration algorithm, $\epsilon$-greedy, to the existing training algorithm, allowing ASNet to take advantage of exploration. We implemented a system that uses Expressive Numeric Heuristic Search Planner (ENHSP) (Scala, Haslum, and Thiébaux 2016) — which is one of the most popular numeric planners and is based on heuristic search — for training ASNet as the *teacher planner*. We also optimize the training algorithm with action generation techniques, objective functions, and variations of optimizers. We analyze our approach through an extensive experimental study to evaluate the effectiveness of our planner. Overall, the results show that our approach is effective for a wide variety of problems though we can struggle with problems with large numbers or broad ranges of numeric variables. We believe this work is particularly relevant for those involved in heuristic search, as it expands the scope of heuristic-search-based planners to encompass the generation of generalized policies for numeric planning problems.

## Background

In this section, we provide background on classical, numeric, and generalized planning and describe the Action

Schema Network approach, which forms the basis of our system.

## Classical, Numeric, and Generalized Planning

Automated Planning involves finding a sequence of actions that lead to a desired goal from a given initial state of a problem. This paper is concerned with two types of planning problems: *classical* and *numeric*. In classical planning, problems are modelled using only propositions (*i.e.* boolean variables) called *fluents*. However, complex real-world problems often require resources to be assigned, consumed, or have a cost defined. These features cannot always be represented using only boolean variables. This has led to the introduction of *numeric planning*, which allows for numerical fluents. Solving a numeric planning problem requires reaching a state where both a given set of propositions hold and a set of numeric constraints are satisfied.

A numeric planning problem can be formally defined as a tuple $P = <F_p, F_n, O, I, G>$, where $F_p$ is the set of boolean propositional variables, $F_n$ is the set of numeric variables (each with its own domain), $O$ is the set of actions, $I$ is the initial state, and $G$ is the set of goal conditions. The goal conditions (or action preconditions) can be either propositional or numeric. A propositional condition is a positive literal (*i.e.* requiring that a particular fluent is true in the corresponding state), while a numeric condition is defined as $c : \xi \triangleright 0$, where $\triangleright \in \{=, \neq, <, \leq, \geq, >\}$ is a relational operator, and $\xi$ is an arithmetic expression over the numeric fluents $F_n$ and the rational numbers $\mathbb{Q}$. For example, a condition may require that $f_1 + 2f_2 - 5 < 0$ must be true for fluents $f_1$ and $f_2$.

The actions $a \in O$ are defined by a set of preconditions and effects. The preconditions determine when $a$ is applicable: they include the set of propositions that must be true in the current state and a set of numeric conditions that the numeric fluents must satisfy. The effects of an action define how the fluents change after the action is applied. This includes which set of propositional fluents change their truth value and assign new numeric values to some subset of the numeric variables.

*Expressive Numeric Heuristic Search Planner (ENHSP)* (Scala, Haslum, and Thiébaux 2016) is a popular numeric planner that we use extensively in our system. At its core, ENHSP is a heuristic forward search planner that can use a variety of domain-independent heuristic functions and search techniques to solve numeric planning instances. In our work, we use ENHSP-20 (Enrico Scala 2020), which is the latest version of this planner, and we configure it to run the A* algorithm guided by the numeric heuristic $opt-h^{max}$ (Scala, Haslum, and Thiébaux 2016). This ensures that any plan returned by ENHSP is optimal.

To ease the process of modelling, classical planning problems are typically encoded as sets of *lifted propositions*, *action schemas*, and *objects*, in a language such as PDDL (Garrido, Fox, and Long 2002), which are then passed to a planner. A lifted proposition is a predicate with input parameters, whose value can be one of the given objects (*i.e.* the free parameter `?x` of the predicate `in(?x)` may be the object `roomA`). An *action schema* is a *lifted* representation of the actions that allow for input parameters (*i.e.* the action `move(?p1, ?p2)` to move from location `?p1` to `?p2`). Action schemas and lifted propositions can then be *ground* by the planner so that it fits the formal definition given above by computing all valid instantiations that assign objects to the arguments of proposition and action parameters. For example, for objects `roomA` and `roomB`, we can generate ground action `move(roomA, roomB)` which has `in(roomA)` as a precondition. Since many numeric variables can take on infinite possible values, numeric planning problems can not be grounded. We describe our approach to this issue below.

*Generalized planning* is a variant of classical planning in which the objective is to find a *policy* that can solve any problem instance from the same domain (*i.e.* planning problems that all share the same action schema and set of lifted propositions). A policy $\pi$ is a mapping from a state to a distribution over the set of actions[1]. While this is similar to the objective of reinforcement learning agents, we note the significant difference in that generalized planners are also given domain information about the action schema and fluents in the domain.

## Action Schema Networks

In this work, we consider generalized planning for numeric planning problems. Our planner is based on *ASNets* (Toyer et al. 2020), a neural network based generalized planning approach for classical planning. Below we describe the structure of an ASNet and how they are trained.

**The Architecture of an ASNet**   An ASNet takes in features about the current state and goal and outputs a probability distribution over the actions applicable in that state (*i.e.* the network encodes a policy). The general structure can be seen in Figure 1, which shows that an ASNet consists of alternating layers of action and proposition modules. Each module is a single network neuron with one for each proposition and action in the domain. These modules are trained using a given set of training problems; thus, this approach assumes that for solving a generalized problem domain, we are also given a set of problems for this purpose.

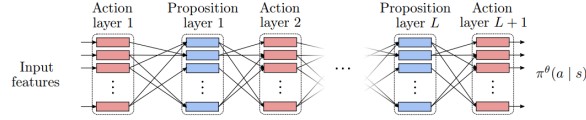

Figure 1: ASNet Structure. Image from Toyer et al. 2020.

For each new problem instance, a new ASNet is constructed from the previously learned modules. This means that the weights of the networks for different problems are shared. This is because, for different problem instances, the grounded actions and propositions vary, meaning a different

---

[1]Some generalized planning systems try to find a *generalized plan*, which is a single sequence of actions that can solve any problem in the domain. We focus on the variant which finds policies in this work.

network is created specifically using the grounded propositions and actions in a problem instance. As such, different modules will often be connected differently in the networks for different problems.

While we leave a full description of ASNets to the original paper (Toyer et al. 2020), we briefly describe their structure and how they are trained here. To do so, we must first define the notion of relatedness. An action and proposition are said to be *related* when the proposition appears in one of the action's preconditions or effects.

The input to an ASNet consists of three bit-vectors. The first vector identifies which of the propositions related to an action $a$ is true in the current state. The second vector identifies which of the propositions related to $a$ is true in the goal state. The final part is a single bit indicating whether or not $a$ can be applied in the current state. These three parts are computed and given as input to each action module on the first level.

For every level after the first, the input of each action module is the output of modules corresponding to the related propositions at the previous level. The size of each module's output is called the *hidden representation size* and is typically set to 16 as in the original ASNet work (Toyer et al. 2020). Thus, if there are 3 related propositions for action $a$, it will take in $3 \cdot 16$ values. The size of the output of each module (including the first-level modules but not the last) is also the hidden representation size. Proposition modules are defined analogously in that their input is given by the output of the modules of the related actions in the previous level[2].

Finally, the output of each action module in the last level is a single real value. This output can be viewed as the relative value of taking the given action, where higher is better. It can be used in several ways to generate a policy. When testing the policy of an ASNet, the original paper selects actions greedily. However, the learning objective for updating neural network weights assumes that the probability of selecting a given applicable action, namely $\pi(a|s)$, is given by taking a softmax over the value of all applicable actions.

**Training an ASNet**  Let us now consider how an ASNet is trained for classical planning problems. The input to the training procedure is a set of fluents, an action schema, a set of training problems $P_{train}$ (*i.e.* start states and goal conditions) that use only those fluents, and the number of epochs to run. The output is a trained set of action and proposition modules. The training begins by generating all action and proposition modules and initializing the neural network weights using Xavier initialization (Glorot and Bengio 2010).

Every training problem is used in the following two-step procedure during every epoch of training. In the first phase, called *exploration phase*, the ASNet for a problem $p \in P_{train}$ is generated, and the current policy is used to generate a sequence of actions of up to $N$ actions (for some

hyperparameter $N$) that will either solve the problem or not. This sequence of actions will generate a set of additional states $s_0, ..., s_N$. For each $s_i$, a *teacher* planner is used to find a plan from $s_i$ to the goal. This is called a *teacher-rollout*. The teacher planner is typically a standard (*i.e.* non-generalized) classical planner. Intuitively, the teacher planner provides plans that the ASNet should mimic, so they will be used to update the network weights as described below. In the original ASNet work, Fast Downward (Helmert 2006) is used as the teacher planner.

The exploration phase is repeated $T_{explore}$ times, where $T_{explore}$ is a hyperparameter. This means that $T_{explore}$ trajectories per training example are generated per epoch. The states along each plan generated by the teacher planner are also added to a state memory buffer $M$, which is maintained across epochs. This buffer stores the states along all paths the teacher planner finds and the action the teacher planner takes in each.

The state memory $M$ is key for the *learning phase* of each epoch of the training procedure. For a given hyperparameter $T_{train}$, this phase is repeated $T_{train}$ times per epoch. Each time it is invoked, a minibatch of examples is sampled from $M$. This minibatch is then used to update the network's weight using a form of *imitation learning*. More specifically, the weights are updated to minimize *cross-entropy loss* between the network's policy generated using the softmax over the outputs of the modules in the last layer. Intuitively, this encourages the network to increase the probability it selects the same action as the teacher planner for the states sampled from the minibatch.

## Action Schema Networks for Numeric Planning

In this section, we explain how we apply ASNets in the context of numeric planning. This includes describing how we convert a numeric problem to a classical planning problem, changes to the neural network structures, and updates to the training procedure.

### Propositionalizing a Numeric Problem for ASNets

To allow ASNets to be used for numeric planning problems, we constructed a propositional approximation of the numeric fluents. Once that is done, the resulting problem has only propositional fluents and thus can be handled by an ASNet. We describe the process for doing so below.

In its original form, ASNet uses a library called MDP-Sim (Younes et al. 2005) to ground the given problems, but this library can only handle boolean fluents. As such, we use ENHSP as part of the process and augment it in order to construct problems suitable for ASNet. First, we use ENHSP to extract the given domain's lifted propositions and action schema. We also use ENHSP to extract the set of objects from all the given training problems, as these are assumed to yield the set of all possible objects in the domain [3].

Next, we propositionalize the numeric variables. Since numeric variables can take on an infinite number of values,

---

[2]Because the set of actions applicable may be different for different problems, there is an additional pooling step so that the same proposition modules can be used across problems. We leave a full description of this step to the original ASNet paper (Toyer et al. 2020)

---

[3]This could be extended to take in a set of objects that is a superset of the objects in the problem files if needed.

most of which are not needed for most problems, we identify a suitable range for each variable, discretize that range, and set a propositional fluent for each value in this discretization. That is, for each numeric proposition $m \in F_n$, we create a set of propositions $< p_0, p_2, ..., p_k >$, where $p_0$ corresponds to $m$ taking on the minimum value $Min$ of its range, $p_k$ corresponds to $m$ taking on the maximum value $Max$ of its range, and for each $0 < i < k$, $p_i$ corresponds to $m$ taking on the value $Min + i \cdot \Delta$ where $\Delta$ is a step size hyperparameter.

In our current system, selecting the range for each numeric variable is done manually by inspecting the domain specification and selecting reasonable values. This is straightforward for most of the standard benchmark domains, as the numeric variables already only take on values along a grid-like space. While this is currently a limitation of our system, we believe this process can be effectively automated using information from plans found by the teacher planner on the training problems. We leave such an investigation as future work.

Once the numeric fluents have been propositionalized, the lifted propositions and action schemas are grounded. This largely follows the existing approach, except the numeric components of the action preconditions and effects are replaced by the disjunction of the sets of the new propositions that ensure the original conditions are satisfied. For example, if an action requires that `fuel >= 10` in its numeric form, then the action will be applicable after it is propositionalized if `fuel_10` is true or `fuel_11` is true or `fuel_12` is true, etc.

This approach of propositionalizing the numeric problem allows ASNets to be used directly as originally described. We have done so in our system, though we have adjusted the training approach described below. An advantage of this approach is that if ASNets are improved in the future, they can just as easily be applied to numeric problems. However, there are also several disadvantages. The first is that the resulting classical problem only approximates the original numeric problem. While we found this effective for most benchmarks, this will not extend to all possible problems. We also found that this greatly increased the number of propositions related to any action. For example, for the fuel example above, all of the propositions which may make the action true are now related to the action. The resulting number of actions and modules (specifically the propositional ones) was thus significantly higher than those in standard classical benchmarks.

### The Updated Training Procedure

In this section, we describe how our training procedure differs from the original ASNet work. The overall approach is very similar, though with a few key differences. First, at the end of each epoch, we test the effectiveness of the learned neural network modules on all problems in the training set by using the greedy policy to solve each. The training terminates if the resulting ASNet exhibits stability over all of the problems for 15 consecutive epochs.

The second main difference is in the policy action selection used to generate trajectories from each training instance as part of the *Explore-Trajectories* function. In the original ASNet work, actions were selected greedily according to the last-layer output from amongst the applicable actions. In our work, we also tested selecting actions using an $\epsilon$-greedy approach (Sutton and Barto 1998). On every step, this approach will select actions greedily with probability $(1 - \epsilon)$, where $\epsilon$ is a hyperparameter, and otherwise selects an action uniformly at random from amongst the applicable actions. This was found to improve performance by forcing the planner to consider a more varied set of states.

The exploration phase consists of the steps explained in the previous section, with a few additions we will now describe. The function *Explore-Trajectories* is performed for each problem in our training set $P_{train}$ for *target-rollouts-per-epoch* number of times. This value refers to how many trajectories we want to explore in each epoch. For classical planning problems, *target-rollouts-per-epoch* was set to 100, i.e. the number of trajectories observed in each epoch would be 100. Numeric planning problems take much longer for each problem instance to run, so exploring too many trajectories is not feasible. We have reduced this number to a lower value as given in the experiments.

Finally, we use the numeric planner ENHSP (Scala, Haslum, and Thiébaux 2016) to perform the teacher rollout. In particular, we set ENHSP to use the $opt - h^{max}$ heuristic, which ensures that all plans it returns are optimal. This optimal plan is then used to add states to the memory buffer $M$ as described in the section titled "Background".

### Implementation

The original ASNet code was developed in Python, using Tensorflow for neural network operations and training. ENHSP-20 is written in Java. Thus, we built a bridge using the Jpype software package to allow the two systems to interact. A string representation of the states and actions are then passed between ENHSP and ASNet to allow for teacher rollout requests and the resulting plans, and also to help ground the numeric fluents.

## Experiments

In this section, we describe several experiments to evaluate the performance of ASNet on numeric planning domains. This includes an evaluation of in-sample performance on training problems and the performance of the policy on unseen problems, a look at training and runtime, and an examination of how well the method scales with problem size.

### Experimental Setup

All the experiments were trained and evaluated on a machine equipped with 64GB of memory and an x86-64 processor clocked at 2.2GHz. We used problems from 8 different domains for our evaluation. Gripper-Tray(Kuroiwa, Shleyfman, and Beck 2022), Gripper-Simple(Kuroiwa, Shleyfman, and Beck 2022), Plant-Watering (Orig.)(Scala, Haslum, and Thiébaux 2016), Sailing(Orig.)(Scala, Haslum, and Thiébaux 2016), and Fn-Counters(Scala, Haslum, and Thiébaux 2016) are standard benchmarks. To further understand our system, we modified 3 of these domains to gen-

erate the following domains— Plant-Watering (Simp.), Sailing (Simp.) and Numerical Gripper — to allow for closer inspection. The numerical Gripper is a modified version of the Gripper classical planning domain. In Numerical Gripper, instead of navigating through rooms, the robot can move in a linear path from point 0 to 10. Its objective remains the same: transporting and retrieving the ball while following this linear trajectory. The simplification of the domains involved removing some actions and limiting the range of integers.

We evaluate the quality of policies using several metrics. These include the learning time, the length of the plans, as well as the success rate and ideal path rate of the learned policies. For policy $\pi$ and problem $p$, the success rate is the probability that $\pi$ will solve $p$. If $\pi$ is deterministic, this is simply 1 if $\pi$ solves $p$, and 0 otherwise. If $\pi$ is stochastic, we estimate the success rate by sampling plans multiple times on each problem and recording the rate that solutions are found.

The *ideal path rate* of policy $\pi$ on a single problem $p$ measures how closely a deterministic policy $\pi$ mimics the teacher planner. To calculate this metric, we first use the teacher planner to generate a solution to $p$. This will be a sequence of actions $< a_0, ..., a_{n-1} >$ that visits states $< s_0, ..., s_n >$. Next, we use $\pi$ to generate an action $a_i^\pi$ for each $s_i$. The ideal path rate is then given by the number of times that $a_i^\pi$ is equal to $a_i$, divided by $n$.

To select effective hyperparameters for our system, we performed a grid search. The number of layers (2), hidden representation size (16), and non-linear activation function (ELU(Clevert, Unterthiner, and Hochreiter 2016)) were kept constant and set the same as in the original ASNet work (Toyer et al. 2020). The number of epochs for training was kept at 50, with the trajectory length limit as 100 for both training and evaluation. The target rollouts per epoch were kept to 5.

The main grid search was over different values for the learning rate, optimizer, and dropout hyperparameters. No single set of hyperparameters was found to be dominant on all domains, however, the use of the Adam optimizer with a learning rate of 0.00005 and a dropout of 0.25 on the outputs of each layer except the last was effective in most cases. Thus, we use those values in the remainder of this section.

## Training Performance

Figures 2a and 2b show how the success and ideal path rates progress over time on the given training problems. For success rate, Figure 2a shows that in most domains, there is generally an increasing rate trend. In most cases, the success rate reached is high, demonstrating that the ASNet is successfully able to represent a policy for numeric domains that can solve the training problems.

A 100% success rate was reached on Fn-Counters, Plant-Watering (Simp.), Gripper-Simple, and Gripper-Tray. In these cases, learning was terminated early because the network had stabilized. While a perfect success rate was not reached in the Sailing (Orig.) and Sailing (Simp.) domains, our numeric ASNet does very well in the simple sailing domain but not the original complex version. Although our

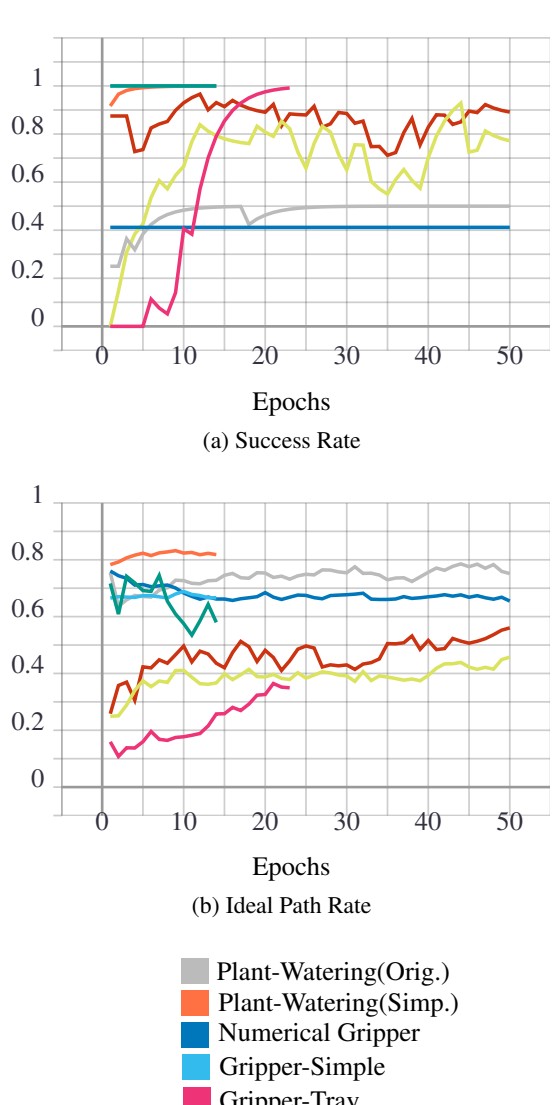

(a) Success Rate

(b) Ideal Path Rate

Plant-Watering(Orig.)
Plant-Watering(Simp.)
Numerical Gripper
Gripper-Simple
Gripper-Tray
Sailing(Orig.)
Sailing (Simp.)
Fn-Counters

(c) Legend

Figure 2: Success and Ideal Path Rates for All Domains

method does improve over time in this more complex domain, it struggles to handle all the problems effectively.

Our system also struggles with the Numerical Gripper domain. Our analysis suggests that the system gets stuck in a local minimum where it perfectly handles 40% of the problems after the first epoch of training but cannot extend the policy to handle other problems.

Figure 2b shows that for most of the domains, the ideal path rate largely stays stagnant in most domains. However, the domains which saw the success rate increase slowly, generally also see increases in ideal path rate. Gripper-Tray, Sailing (Orig.), and Sailing (Simp.) are clear examples of this behaviour.

Interestingly, Figure 2b shows that even in the four domains with a 100% success rate — namely Fn-Counters, Plant-Watering (Simp.), Gripper-Simple, and Gripper-Tray — the ASNet policy is not necessarily mimicking the teacher planner. Instead, the planner is partially following the teacher planner, but is finding a policy that is easier to learn.

## Generalizing to New Problems

In our previous experiment, we noticed that one cause of failure for ASNet was that the standard policy which selects greedily on the output of the last layer would sometimes get stuck in loops. In the simplest such case, the policy would get stuck going back and forth between the same two states. To remedy this situation, we also tested the ASNet by using a stochastic policy. This policy corresponds to the learning objective: the probability of taking an applicable action is given by the softmax over the output of the last layer of the ASNet. To estimate the success rate of this policy, we average over 20 runs per problem. In this experiment, for each domain, we tested on 4 test problems that the trained model had not seen before for both greedy and stochastic action selection policies. The size of these test problems was varied with some problems at the same size as the training and some larger than the training. As an illustration, in the Gripper (Simp.) domain, we conducted training with a range of ball quantities, specifically between 4 and 10. On the other hand, the test problem instances in the Gripper (Simp.) domain were diversified by varying the number of balls between 6 and 12. The results are as shown in Table 1. The table also shows the success rate of a uniformly random policy as a baseline. The highest success rate per domain is shown in bold.

We first note that stochastic and greedy solve all the Fn-counters and Numerical Gripper problems. In the Sailing and Plant-Watering variants, the stochastic policy did significantly better than the greedy. As stated above, this was due to the policy running into loops. We have mainly observed this in domains such as these four when there are *move* actions. This trend of going back and forth between 2 actions is broken when the stochastic policy is used, allowing our numeric ASNet to solve the problem instance with a greater probability.

The greedy policy does perform better in Gripper-Simple and Gripper-Tray. We believe the reason for this is that the greedy policy already has a perfect success rate in these domains, so adding stochasticity can only hurt. We note that this does suggest an interesting approach to find plans in practice: first, run the greedy policy and then turn to using a stochastic policy if the greedy policy fails. This will have the benefit of both approaches.

Finally, we note that the learnt policies are significantly outperforming the purely random policy in all cases except Fn-counters in which case all methods have 100% coverage. This clearly demonstrates our numeric ASNet is learning useful information about the domain that can be used to solve new problems.

| Domain | Action Selection Policy | | |
| --- | --- | --- | --- |
| | Stochastic | Greedy | Random |
| Plant-Watering (Original) | **0.55** | 0.00 | 0.01 |
| Plant-Watering (Simple) | **1.00** | 0.50 | 0.75 |
| Numerical Gripper | **1.00** | **1.00** | 0.06 |
| Gripper-Simple | 0.98 | **1.00** | 0.00 |
| Gripper-Tray | 0.34 | **1.00** | 0.14 |
| Sailing (Original) | **0.48** | 0.25 | 0.00 |
| Sailing (Simple) | **0.62** | 0.50 | 0.20 |
| Fn-Counters | **1.00** | **1.00** | **1.00** |

Table 1: Success rate with different policies on unseen problems.

## Scaling Tests Using Fn-Counters

In this section, we study how our numeric ASNet approach scales to larger problems by focusing on the Fn-counters domain. In Fn-counters, there are two or more "counters", which each take on a positive integer value. The available actions increment or decrement one of the counters at a time. Given some initial values for the counters, the objective is to have the counters take on values in a specific consecutive sequence. For example, suppose there are two counters $c0$ and $c1$ with initial values $c0 = 3$ and $c1 = 1$. Then a typical goal would be that $(c0 + 1) \leq c1$. For this problem instance, the optimal plan would be to apply the increment action three times on the $c1$ counter. The final value of counter $c0$ would be 3 and $c1$ would be 4. An important property of this domain is that if a counter has a value of 0, it cannot be decremented. Similarly, there is a `max_int` fluent which determines the maximum value of any counter, at which point it cannot be incremented.

For this experiment, we set the step size parameter to 1, but considered different `max_int` values to test how well our system scales with an increasing number of possible fluents after the propositionalizing step. In particular, we train our neural network using five problems of 2 counters each. The `max_int` values used were 10, 50, 100, 300, 400, 500, and 1000. The same hyperparameters were used for all experiments.

Our evaluation showed that when `max_int` gets too high, our numeric ASNet fails because it is unable to handle the increase in the number of propositions. In particular, we found that our approach was unable to handle the problems for a value of 400 or above. This is because the number of propositions increases with the size of the problem. In our numeric ASNet, increasing `max_int` also increases the number of propositions related to any action, which can weaken the signal passed on between layers of the network.

Somewhat surprisingly, we found our system found a suboptimal policy for `max_int` values of 10 and 50, but an optimal policy for 100 and 300. To see what is learned when `max_int` is set to 10, consider the problem instance given above where $c0 = 3$ and $c1 = 1$, which can be solved by simply incrementing $c1$ three times. In contrast, the policy ASNet learns increments $c0$ seven times (so that $c0$ is equal to `max_int`), then increments $c1$ nine times (so that $c1$ is

equal to `max_int`), then finally decrements $c0$ once. The network uses this policy of incrementing both counters to the `max_int` value, regardless of the start state. Notice that this means that policy is pushing towards a specific goal state in doing so (namely $c0 = 9$ and $c1 = 10$), instead of any other goal state with $(c0 + 1) \le c1$.

Our hypothesis is that this policy is easier to find and represent using the ASNet architecture, but that it is in a local minimum in weight space that the learning is unable to overcome even given additional time. However, due to another hyperparameter, *trajectory length limit*, this policy is not valid when `max_int` is 100 or 300. This hyperparameter limits the path length of any planning instance being considered during training and is set to 100 by default. However, the suboptimal policy takes more than 100 steps when `max_int` is 100 or 300, so the ASNet is forced to find a better policy. In this case, it finds the optimal one.

We note that this means that the ASNet is capable of representing and finding the optimal policy for the smaller `max_int` values, even though it doesn't. It also suggests that this approach is susceptible to getting stuck in local minima in policy space. We believe that finding ways to combat this issue is an interesting possible area for future work.

### Training and Run-Time Performance

We ran an experiment to compare the time it takes to train all domains to better understand how domain complexity affects training time. To do so, We trained our numeric ASNet on two problem instances of each domain. The hyperparameters were also kept constant across domains. The experiment shows that Fn-counters, with `max_int` of 10 and thus a total of 20 numeric propositions, is the fastest to train, with a time of 28sec. In contrast, Sailing(Orig.) is the slowest to train, with a total of 2800 numeric propositions and a time of 812sec. While factors like the time needed to perform teacher rollouts also matters, these results suggest that training time is typically proportional to the number of numeric propositions in that domain.

We also experimented with evaluating the time it takes to execute a trained model to find a plan for unseen problem instances. Of note, a significant portion of this runtime is in the construction of the ASNet from the trained modules. To see this, we compared the time needed to find a single plan to a new problem to the average time per plan if we run ASNet 20 times on the same problem once it has been constructed. The results are shown in Figure 3a, which shows the first plan takes much longer to generate than the subsequent plans since the ASNet does not need to be recreated each time.

In the next experiment, we compare the time it takes to solve ten new unseen problem instances using a trained numeric ASNet compared to ENHSP. The unseen problem instances would have different initial state values for the objects. These new problems are generated with random integers for the numeric propositions. Figure 3b shows the results of this comparison between ENHSP and the average of over 20 runs per problem when using ASNet. The standard deviation over the problems is also shown in the figure.

The results show that in most cases, ENHSP takes more than double the time to find a plan. However, we note that av-

eraging over 20 runs lessens the impact on the network creation time. When that is included, the runtime between the two planners is comparable. However, the difference when considering the average time over 20 runs demonstrates the potential in generalized planners of this type, as there are significant runtime improvement opportunities to be made simply by improving the process of network creation. It also demonstrates that ASNet can be used quickly and effectively to generate a set of different plans for the same problem (by using stochastic action selection), whereas ENHSP will require significant changes, likely without the runtime benefits, to be used in that way.

### Related Work

Significant research has been conducted on numeric planners (Scala, Haslum, and Thiébaux 2016; Hoffmann 2003) and generalized planners (Toyer et al. 2020). However, there has been limited exploration of generalized planners specifically designed for numeric problems. One notable exception is the generalized linear integer numeric planner (GLINP) (Lin et al. 2022). GLINP requires that all numeric variables are integer ones. A similar approach is taken by Qualitative Numeric Planning (QNP) (Srivastava, Immerman, and Zilberstein 2011) which requires that the effects of actions decrease or increase the value of some variables by an unspecified amount. Our approach shares similarities with GLINP and QNP in that the values of our numeric propositions vary by a consistent step size.

The focus of these systems is on finding more tractable subsets of the generalized numeric planning problem. To this end, QNP only considers goal conditions of the form $v > 0$ or $v = 0$ where $v$ is a numeric variable. GLINP also focuses on the case where the goal is to decrease the value of the numeric variables in the problem. Since these restrictions do not apply to our approach, we did not consider these methods directly comparable. Furthermore, our approach is the first to incorporate neural network-based policy learning into the domain of generalized numeric planning.

### Conclusion

In this paper, we use a grounding process that takes the numeric values possible for each variable and converts it into a fixed range of values. This method propositionalizes the numeric values and structures them in a way that can be input to an ASNet. The training of ASNet has been enhanced by adding the $\epsilon$-greedy method for more variability in the paths, leading to an improved ability to solve unseen problems. We have also added ENHSP as the teacher planner, which provides the ASNet with guidance on how to solve numeric planning problems. The plan returned by ENHSP is used to calculate the plan cost and assign state values which are then used in the learning phase of training to improve the policy of our numeric ASNet. The result is the first generalized planner for numeric planning problems which uses neural networks to learn a generalized policy from training examples.

Our experiments show that applying a trained ASNet on new problem instances from a domain and executing the

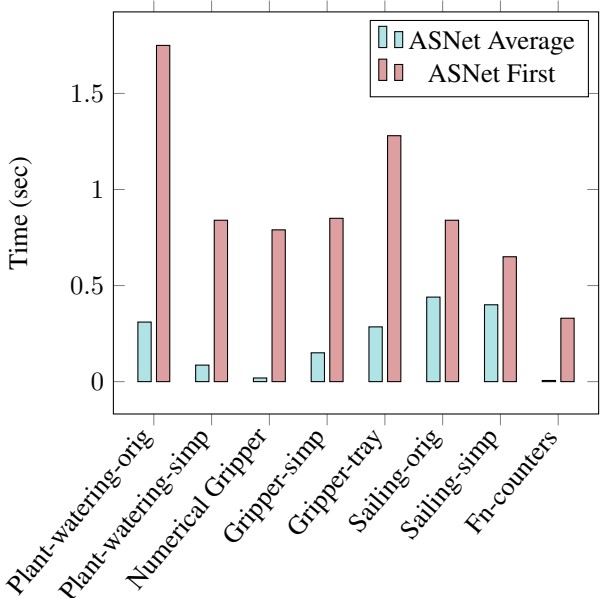

(a) ASNet execution time

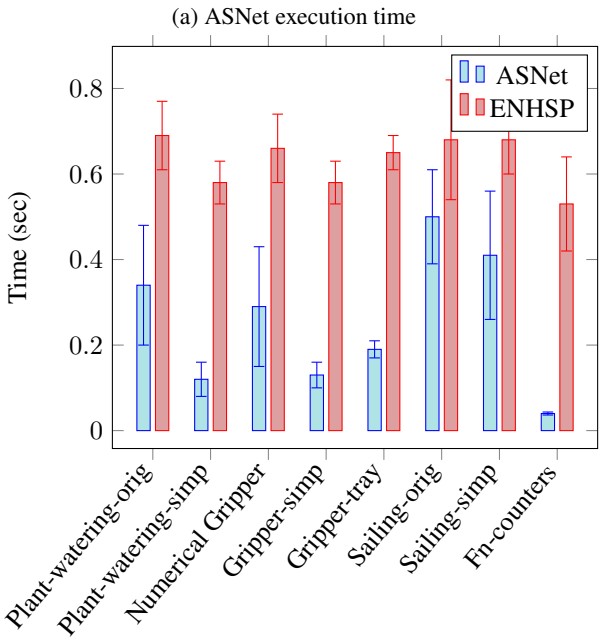

(b) Comparing ASNet to ENHSP.

Figure 3: An analysis of the time to generate plans using ASNet and ENHSP.

same problem instances using the heuristic search planner, ENHSP, results in much faster execution using our numeric ASNet. This is because ASNet has already learned the policy and only needs to apply that to the problem instances. In contrast, the baseline planner cannot transfer the learnt knowledge from one problem instance to another and so ends up having to re-evaluate every problem instance from scratch. This demonstrates that the promise of generalized planning carries over to the numeric case.

Human pre-processing for the grounding technique is necessary when assigning bounds to the variables which is a current limitation of this method. This method has been shown to significantly increase the input size when the range of the numeric variables increases or the step size decreases. This increase affects the performance of our numeric ASNet and can potentially lead to a memory problem if the numeric values are too large to store and handle. However, we view our work as an important first step to bridging the gap between generalized numeric planning and neural networks. In future work, we aim to improve the network construction process and develop an approach that directly takes the numeric values of inputs to the ASNet, thus avoiding the current system's limitations.

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
