# OpenReview forum: "Action Schema Networks for Numeric Planning"
_icaps-conference.org/ICAPS/2023/Workshop/HSDIP — ICAPS HSDIP 2023_

### Official Review · Reviewer_SykP · 2023-04-12
**Limited relevance to the workshop**

**Rating:** 5
**Confidence:** 4

**Review:**

The paper deals with extending the ASNets to numeric planning, to create a policy for a problem in a domain. As in the original work, it is done by learning weights to be used in order to define a network per given problem. The extension is straightforward, via discretization of the numeric fluents.

The paper does not deal with issues in heuristic search or heuristic computations and does not seem to hit any of the topics of interest of the workshop. I am afraid that it might be of limited interest to the workshop participants. For that reason, and with a heavy heart, I could not currently recommend acceptance, unless the authors can present convincing arguments for the relevance to the workshop.

Minor (or possibly major, I am not a numeric planning person) comment: as far as I know, it's numeric planning, not numerical planning.

---

> ### Author Response · Authors · 2023-04-30
> **Response for Review**
>
> Thank you for your review. We will update references from numerical to numeric planning.
>
> Regarding the relevance to the workshop, we believe the work is of interest to those working on heuristic search since it increases the applicability of heuristic-search based planners to include the generation of generalized policies for numeric planning problems.

---

### Official Review · Reviewer_vvk9 · 2023-04-26
**Learning from planners to develop policies**

**Rating:** 5
**Confidence:** 4

**Review:**

Summary:

This paper presents an adaptation of Action Schema Networks to work with numerical planning problems. It uses heuristic search planners to train the model, and then applies policies to both seen and unseen problem instances. The authors demonstrate that the policies learned are often effective and novel in relation to the planners used to train, but the training can be subject to local minima.


Specific Feedback:

The approach is straightforward and the authors demonstrate good results for the problems examined. This direction could be useful for numerical planning, but as there is no heuristic learned or any search-based planning being done beyond the training period, the fit for this workshop is poor.

The analysis of results provides interesting guidance for handling the problems that arose, such as loops encountered by the greedy policy being solved by falling back to a stochastic policy.

Because the model is trained by an existing planner, it seems that the training must be done on only small problems. The authors show results for policies on unseen problems, but don't discuss whether those unseen problems are of the same size as the originals. Can the policies learned on small problems be applied to larger instances? If so, splitting out the results to show if there is a loss of quality as the problems grow would help demonstrate the importance of this approach.


Minor points:

Several hyperparameters are mentioned at various points. A list of all hyperparameters in one clear place would be helpful for clarity.

Paragraph that begins "To ease the process": missing space after "PDDL"

Under "The Updated Training Procedure": missing reference in "summarized in section ."

"Figure 2b shows that for most domains..." sentence has repetition of "in most domains".

---

> ### Author Response · Authors · 2023-04-30
> **Response to Review**
>
> Thank you for your review. We will fix the other grammatical and stylistic changes suggested.
>
> Regarding the question about the size of the problems, we have varied the size of the problems, so some problems are the same size as the training problems, and some are larger than the training problems.
>
> For the relevance to the workshop, we believe the work is of interest to those working on heuristic search since it increases the
> applicability of heuristic-search based planners to include the generation of generalized policies for numeric planning problems.

---

### Decision · Program_Chairs · 2023-05-06

**Decision:**

Accept

**Comment:**

We are happy to announce that the paper is accepted to the workshop.

For the final version, we ask that you address the concerns and corrections mentioned by the reviewers, especially anything you can do to emphasize the connections to heuristic search planning and ways that this work can advance the field of heuristic search.